



# The role of atmospheric $CO_2$ in controlling patterns of sea surface temperature change during the Pliocene

Lauren E. Burton[1], Alan M. Haywood[1], Julia C. Tindall[1], Aisling M. Dolan[1], Daniel J. Hill[1], Erin L. McClymont[2], Sze Ling Ho[3], and Heather L. Ford[4]

[1]School of Earth and Environment, University of Leeds, Leeds, West Yorkshire, LS2 9JT, UK
[2]Department of Geography, Durham University, Durham, DH1 3LE, UK
[3]Institute of Oceanography, National Taiwan University, Taipei, Taiwan
[4]School of Geography, Queen Mary University of London, London, UK

*Correspondence to*: Lauren E. Burton (eeleb@leeds.ac.uk)

**Abstract.** We present the role of $CO_2$ forcing in controlling patterns of Late Pliocene sea surface temperature (SST) using seven models from Phase 2 of the Pliocene Model Intercomparison Project (PlioMIP2) and palaeoclimate proxy data from the PlioVAR working group. At a global scale, SST change in the Late Pliocene relative to the pre-industrial is predominantly driven by $CO_2$ forcing in the low and mid-latitudes and non-$CO_2$ forcing in the high latitudes. We find that $CO_2$ is the dominant driver of SST change at the vast majority of proxy data sites assessed (17 out of 19), but the relative

dominance of this forcing varies between all proxy sites, with $CO_2$ forcing accounting for between 27% and 82% of the total change seen. The dearth of proxy data sites in the high latitudes means that only two sites assessed here are predominantly forced by non-$CO_2$ forcing (such as changes to ice sheets and orography), both of which are in the North Atlantic Ocean.
We extend the analysis to show the seasonal patterns of SST change and its drivers at a global scale and at a site-specific level for three chosen proxy data sites. We also present a new estimate of Late Pliocene climate sensitivity using site-specific

proxy data values. This is the first assessment of site-specific drivers of SST change in the Late Pliocene and highlights the strengths of using palaeoclimate proxy data alongside model outputs to further develop our understanding of the Late Pliocene. We use the best-available proxy and model data, but the sample sizes remain limited and the confidence in our results would be improved with greater data availability.

## 1. Introduction

The Late Pliocene (~3.6-2.6 Ma), particularly the mid-Piacenzian Warm Period (mPWP; 3.264-3.025 Ma), is a key focus of palaeoclimate research as one of the best analogues for the near-term future (e.g., Haywood et al., 2011; Burke et al., 2018; Tierney et al., 2020; Forster et al., 2021). The Late Pliocene is well-placed to be considered as an analogue given that it is the most recent period of sustained warmth above pre-industrial (PI) levels, has an atmospheric $CO_2$ concentration elevated above PI levels, and has similar continental configuration to modern.





Over the past 35 years there has been a concerted effort to collate and synthesise disparate geological information from the Late Pliocene to build a progressively more complete spatial picture of the patterns of change. In particular, in the last decade the reconstruction efforts of the modelling and data-model communities have adopted a "time slice" approach and focused on a specific interglacial within the Late Pliocene. Selected as the target for Phase 2 of the Pliocene Model Intercomparison Project (PlioMIP2; Haywood et al., 2016a) and the upcoming Phase 3 (PlioMIP3; Haywood et al., in press),

Marine Isotope Stage (MIS) KM5c is a warm interval with orbital forcing very similar to modern, characterised by a negative benthic oxygen isotope excursion (0.21-0.23‰) centred on 3.205 Ma (Lisiecki and Raymo, 2005; Haywood et al., 2013). KM5c also has an atmospheric $CO_2$ concentration similar to modern, with a central estimate of $371^{+32}_{-29}$ ppm (de la Vega et al., 2020).

In addition to these synthesis efforts, additional proxy data from new sites and at higher temporal resolution mean that we

are beginning to better understand the temporal variability in the Late Pliocene, but understanding the cause of the changes we see in proxy records for a specific site remains difficult. Here, we integrate a novel modelling method with the best available Late Pliocene geological sea surface temperature (SST) data to shed insight on the causes of sea surface temperature patterns during the Late Pliocene.

### 1.1. Synthesis of recent geological data

The U.S. Geological Survey Pliocene Research Interpretation and Synoptic Mapping (PRISM) project has been instrumental in documenting geological data for the Pliocene for over three decades. PRISM reconstructions have been used in both phases of PlioMIP: PlioMIP1, assessing the mPWP, used the PRISM3D reconstruction (Dowsett et al., 2010), while PlioMIP2, assessing the KM5c time slice, used the PRISM4 reconstruction (Dowsett et al., 2016).

Alongside the PRISM project, the Past Global Changes (PAGES) "PlioVAR" working group has also compiled geological

data with a remit of assessing Pliocene climate variability on glacial-interglacial timescales, including the mPWP and the following period of intensified Northern Hemisphere glaciation (McClymont et al., 2017, 2020; McClymont, Ho et al., 2023). The PlioVAR working group developed robust stratigraphic constraints that allowed for a detailed view of ocean temperatures during KM5c (McClymont et al., 2020). Geological proxy data was only included from sites that had either ≤ 10 kyr resolution benthic δ¹⁸O data which could be tied to the LR04 stack (Lisiecki and Raymo, 2005) or the HMM-Stack

(Ahn et al., 2017), or the palaeomagnetic tie points for upper Mammoth (C2An.2n (b) at 3.22 Ma) and lower Mammoth (C2An.3n (t) at 3.33 Ma). Full details on the age models used are presented in McClymont et al. (2020).

Two proxy reconstructions of SST are included in the PlioVAR KM5c synthesis: from alkenones using the $U^{K'}_{37}$ index and from planktonic foraminifera Mg/Ca proxies (McClymont et al., 2020; see also Section 2.3). Mid-Piacenzian SSTs have previously also been reconstructed (e.g., O'Brien et al., 2014; Petrick et al., 2015; Rommerskirchen et al., 2011) using the

TEX₈₆ proxy (Schouten et al., 2002), but these data were not included in the PlioVAR KM5c synthesis as they could not be confidently assigned to the KM5c interval (McClymont et al., 2020). Results from the $U^{K'}_{37}$ and Mg/Ca proxy data were used in tandem and also compared to assess the impact of the choice of proxy for SST reconstruction.

The combined $U_{37}^{K\prime}$ and Mg/Ca proxy data produced a global annual mean SST anomaly of +2.3°C for KM5c relative to the PI, with the largest anomalies in the mid- and high-latitudes and a reduction in the meridional SST gradient of 2.6°C. This

global mean SST warming derived from the two proxies is equal to the warming shown in the PlioMIP2 ensemble mean, with 10 models indicating less warming than this and 6 models indicating more warming (McClymont et al., 2020).

PlioVAR has also examined the climate following KM5c and the onset and intensification of Northern Hemisphere glaciation (McClymont, Ho et al., 2023). An updated planktonic foraminifera Mg/Ca reconstruction was created as part of this analysis, covering the KM5c interval. Assessing Pliocene climate variability on a longer timescale reinforces that

targeting a specific interglacial allows for best data-model comparison efforts due to the minimisation of orbital-scale variability (McClymont, Ho et al., 2023; McClymont et al., 2020; Haywood et al., 2020).

## 1.2. Using climate models to aid interpretation of geological data

Despite the long history of geological data synthesis, understanding the cause of a given climate signal remains challenging. As climate models have developed, their ability to be used synergistically alongside geological proxy data has increased and

there is now a strong precedent for using models to support the interpretation of proxy data (e.g., Salzmann et al., 2008, 2013; Tindall et al., 2017).

There are multiple ways in which this synergistic model-proxy data relationship can be explored. On a basic level, we can compare proxy data to model data to test how well signals of change are reconstructed in a given time period; proxy data and models agreeing on the sign and amplitude of change gives us confidence in both methods and suggests an ability to use

models to explore the drivers and processes behind signals seen in proxy data. Conversely, disagreement between proxy data and models can lead to a decrease in confidence and questions around the cause of different signals (Tindall et al., 2022; see also Haywood et al., 2016b; McClymont et al., 2020). Such disagreements reveal shortcomings and limitations of either the proxies and/or the climate models, and outline avenues to further improve them.

Climate models are also capable of simulating the original proxy signal (e.g., the isotopic signal incorporated into

foraminifera) rather than the calculated temperature using the isotope ratio, which includes additional sources of potential uncertainty in its derivation. In turn, models can help our understanding of what might be controlling the proxy signal in a given time and space. For example, Tindall et al. (2017) use an isotope-enabled version of the HadCM3 model to directly simulate pseudo-coral and pseudo-foraminifera data in the Pacific to explore the expression of El Niño in the Pliocene. By using isotope-enabled models in this way it is possible to see regions where the isotopic expression of ENSO is pronounced

(e.g., the central Pacific (Tindall et al., 2017)), which allows us to assess whether the proxy data signal at specific sites is driven by ENSO or another form of variability.

Palaeoclimate proxy data and modelling outputs have also been used synergistically to constrain estimates of climate sensitivity, particularly for the Late Pliocene (e.g., Hargreaves and Annan, 2016 and references therein; Haywood et al., 2020) and the Last Glacial Maximum (e.g., Renoult et al., 2020). Hargreaves and Annan (2016) present an estimate of

equilibrium climate sensitivity (ECS) of 1.9-3.7°C for the mPWP using PlioMIP1 model output and proxy data from



PRISM3 (Dowsett et al., 2009). Haywood et al. (2020) extends and adapts this analysis for the PlioMIP2 model outputs and generate a site-specific estimate of ECS using the mPWP SST reconstruction of Foley and Dowsett (2019).

Here we present another example of using climate model outputs and geological proxy data synergistically to build a clearer picture of Late Pliocene environmental change, by exploring the dominant cause of SST change at specific proxy data sites.

We apply the "FCO$_2$" method of Burton et al. (2023, detailed below) using outputs from PlioMIP2 to explore the local forcings at individual proxy sites with reference to CO$_2$ forcing and palaeogeographic boundary condition changes. We then discuss the implications of these results, including what the model output may indicate at a seasonal scale that the proxy data cannot resolve (Section 4.1), as well as a new estimate of Late Pliocene climate sensitivity (Section 4.2).

## 2. Methods

**2.1. FCO$_2$ method**

The FCO$_2$ method was first presented in Burton et al. (2023) and shows the proportion of the total Pliocene minus PI climate change that is due to CO$_2$ forcing. The method uses three experiments from PlioMIP2: Eoi$^{400}$, E$^{280}$ and E$^{400}$ (Table 1). At the time of compiling this study, six modelling groups had completed the E$^{400}$ experiment for SST and these six models are used as a subset of the PlioMIP2 ensemble (see Section 2.2).

| Experiment name | Description | Land-sea mask | Topography | Ice | Vegetation | CO$_2$ (ppm) | Status |
|---|---|---|---|---|---|---|---|
| Eoi$^{400}$ | Pliocene control experiment | Pliocene – Modern | Pliocene | Pliocene | Dynamic | 400 | Core |
| E$^{280}$ | PI control experiment | Modern | Modern | Modern | Dynamic | 280 | Core |
| E$^{400}$ | PI experiment with CO$_2$ concentration of 400 ppm | Modern | Modern | Modern | Dynamic | 400 | Tier 2 – Pliocene4Pliocene and Pliocene4Future |

**Table 1: Names and descriptions of the three PlioMIP2 experiments used in the FCO$_2$ method (Burton et al., 2023).**

FCO$_2$ is calculated by:

$$FCO_2 = \frac{(E^{400} - E^{280})}{(Eoi^{400} - E^{280})}$$

where E$^{400}$-E$^{280}$ represents the change in climate caused by the change in CO$_2$ concentration from 280 to 400 ppm alone, and Eoi$^{400}$-E$^{280}$ represents the change in climate as a result of implementing the full Pliocene boundary conditions. Although this

paper focuses on SST change, the FCO$_2$ method can be applied to any climate parameter so long as the necessary model experiments (Table 1) have been run.

The FCO$_2$ calculation typically produces a result between 0 and 1, where 1 represents a change wholly dominated by CO$_2$ forcing and 0 represents the opposite case where change is wholly dominated by non-CO$_2$ forcing. In keeping with the





PlioMIP2 experimental design, non-$CO_2$ forcing is defined as changes to ice sheets and orography, the latter of which also

includes changes to prescribed vegetation, bathymetry, land-sea mask, soils, and lakes (Haywood et al., 2016a).

$FCO_2$ values above 1 and below 0 can occur in rare instances (see Burton et al., 2023). If the effect of $CO_2$ forcing is in the opposite direction to the overall climate signal (i.e., $E^{400}$-$E^{280}$ > 0 but $Eoi^{400}$-$E^{280}$ < 0, or $E^{400}$-$E^{280}$ < 0 but $Eoi^{400}$-$E^{280}$ > 0) then $FCO_2$ will be below 0. If the effect of $CO_2$ forcing is greater than the overall climate signal (i.e., $E^{400}$-$E^{280}$ > $Eoi^{400}$-$E^{280}$ > 0, or $E^{400}$-$E^{280}$ < $Eoi^{400}$-$E^{280}$ < 0) then $FCO_2$ will be above 1. Such values are mostly commonly seen where the $Eoi^{400}$-$E^{280}$

anomaly is small, and the usefulness of the $FCO_2$ method is limited in these cases where there is little climate signal to explain.

The $FCO_2$ method is only suited to quantify the proportion of the total change that is attributable to $CO_2$ forcing or non-$CO_2$ forcing. The method alone cannot expand on the physical mechanisms or processes, though it is possible to make suggestions based on e.g., knowledge of the oceanographic setting at a given proxy site. In order to comment further on the

non-$CO_2$ forcings it would be necessary to complete further forcing factorisation model experiments which is beyond the scope of this paper, but is a suggested target for future modelling work looking towards PlioMIP3.

In this paper, as in Burton et al. (2023), uncertainty in the $FCO_2$ analysis is considered in terms of whether there is consistent agreement between the individual models on whether $CO_2$ forcing ($FCO_2$ > 0.5) or non-$CO_2$ forcing ($FCO_2$ < 0.5) is the most important driver of change. $FCO_2$ is deemed to be uncertain in regions where three or fewer of the six models agree on the

dominant forcing.

## 2.2. Participating models and model boundary conditions

For a model to be included in this study it had to meet the criteria of completing the $Eoi^{400}$, $E^{280}$ and $E^{400}$ experiments for SST, with outputs spun up to equilibrium. Six of the 17 models included in PlioMIP2 met these criteria: CCSM4-UoT, CESM2, COSMOS, HadCM3, MIROC4m and NorESM1-F. The models vary in age and resolution; summary details are

shown in Burton et al. (2023), and full details for the PlioMIP2 ensemble are shown in Haywood et al. (2020). This subset of models is representative of the whole PlioMIP2 ensemble (Table 2). Standardised Pliocene boundary conditions are used in all models in PlioMIP2 – including the six models here – which are derived from the U.S. Geological Survey PRISM4 reconstruction (Dowsett et al., 2016) and implemented as described in Haywood et al. (2016a).

| Parameter | PlioMIP2 ensemble | This ensemble |
|---|---|---|
| ECS (°C) | 3.7 | 3.8 |
| Earth system sensitivity (ESS; °C) | 6.2 | 6.5 |
| ESS to ECS ratio | 1.7 | 1.7 |
| $Eoi^{400}$-$E^{280}$ SST anomaly (°C) | 2.3 | 2.3 |

**Table 2: A comparison of climate parameters between the PlioMIP2 ensemble and the subgroup of PlioMIP2 models used in this**
**study (adapted from Burton et al., 2023; the adaptation reflects the exclusion of the IPSLCM5A2 climate model from this SST-focused ensemble due to limited data availability).**



The boundary conditions include spatially complete gridded datasets at 1° x 1° of latitude-longitude for land-sea distribution,
topography and bathymetry, vegetation, soil, lakes and land ice cover, and all models here used the "enhanced" version
meaning that they include reconstructed changes to the land-sea mask and ocean bathymetry (Haywood et al., 2020). The
Pliocene palaeogeography is similar to modern, except for the closure of the Bering Strait and Canadian Arctic Archipelago;
increased land area in the Maritime Continent; and a West Antarctic Seaway (Haywood et al., 2016a; Dowsett et al., 2016).
The PRISM4 reconstruction also includes dynamic topography and glacial isostatic adjustment to better represent local sea
level (Dowsett et al., 2016). The atmospheric concentration of $CO_2$ is set to 400 ppm in the PlioMIP2 boundary conditions,
with concentrations for all other trace gases set as identical to those in the PI control experiment ($E^{280}$) for each individual
model group (Haywood et al., 2016a).

The ice configuration in the PRISM4 reconstruction is based upon the results from the Pliocene Ice Sheet Modelling
Intercomparison Project (PLISMIP; Dolan et al., 2015). The Greenland ice sheet is confined to high elevations in the eastern
Greenland mountains, covering an area of around 25% of the modern ice sheet (Dolan et al., 2015; Koenig et al., 2015). The
ice coverage over Antarctica is still a source of debate (see Levy et al., 2022), but the PRISM3 reconstruction (Dowsett et al.,
2010) is supported and so retained in the PRISM4 reconstruction (Dowsett et al., 2016). This configuration sees a reduction
in the ice margins in the Wilkes and Aurora basins in eastern Antarctica, while western Antarctica is largely ice free.

## 2.3. Proxy SST data

The temporal focus of this paper is MIS KM5c (3.205 ± 0.01 Ma), the time slice used in PlioMIP2. Details on KM5c, the age
models used as well as the data assigned to KM5c are presented in McClymont et al. (2020). We adopt a multi-proxy
approach using data from the PlioVAR project (McClymont et al., 2020; McClymont, Ho et al., 2023). Two SST proxies are
assessed: the alkenone-derived $U^{K'}_{37}$ index (Prahl and Wakeham, 1987) and foraminifera calcite Mg/Ca (Delaney et al., 1985).
Both proxies have multiple calibrations to modern SST and the impact of calibration choice on SST data is discussed in
McClymont et al. (2020). As for the PlioVAR analyses (McClymont et al., 2020; McClymont, Ho et al., 2023), SST data
were generated using the same calibration for all $U^{K'}_{37}$ and Mg/Ca measurements to minimise the impact of calibration choice
on differences between sites.

As in McClymont et al. (2020), anomalies relative to the PI are calculated using the ERSSTv5 dataset. We focus on the
BAYSPLINE calibration for alkenone-derived $U^{K'}_{37}$ SST data (Tierney and Tingley, 2018) and an updated PlioVAR
calibration for Mg/Ca SST data (McClymont, Ho et al., 2023). Hereafter, "$U^{K'}_{37}$ data" will refer to the BAYSPLINE dataset
presented in McClymont et al. (2020) and "Mg/Ca data" will refer to the Mg/Ca dataset presented in McClymont, Ho et al.
(2023). The choice of calibration does not significantly impact the $FCO_2$ on SST results (see S1 in the Supplement).

Seven $U^{K'}_{37}$ sites presented in McClymont et al. (2020) are not included here as they fall on land in the model Pliocene land-
sea mask, meaning that an $FCO_2$ on SST value cannot be generated. Four of these sites are also in the Benguela upwelling
region, the driving processes for which are not well captured by current climate models. Though $FCO_2$ on SAT is shown to



be comparable to FCO$_2$ on SST outside of the high latitudes in Burton et al. (2023), the decision was made to exclude these
sites as a site-specific comparison could not be made and taking the nearest ocean grid point may not accurately represent the
oceanographic setting of the proxy site.

The KM5c sites considered here have also been analysed for the PRISM3 time interval (3.264-3.025 Ma; Dowsett et al.,
2016). The data for the PRISM3 interval is only considered in the temporal variability analysis (Section 3.3); data for the

PRISM3 interval represent the mean of the entire period rather than a warm peak average, so are well-suited to assess
temporal variability. No data from the PRISM3 interval is available for site U1337, so this site is also excluded from the
analysis for completeness. All data in other sections are solely for KM5c.

Twenty-one proxy sites are considered in total. Of these, 19 proxy sites have data available for KM5c, 15 of which have $U_{37}^{K'}$
data and six of which have Mg/Ca data (sites U1313 and ODP1143 have data from both proxy types). Site ODP999 has only

Mg/Ca data available for KM5c, but Mg/Ca data and $U_{37}^{K'}$ data for the PRISM3 interval. The remaining two sites (DSDP610
and U1307) have $U_{37}^{K'}$ data available for the PRISM3 interval only (with no data available for KM5c) and are included in
Section 3.3 only.

## 3. Results

### 3.1. FCO$_2$ on sea surface temperature

The location of the proxy sites with reference to the multi-model mean (MMM) Eoi$^{400}$-E$^{280}$ SST anomaly and FCO$_2$ on SST
are shown in Fig. 1. The MMM global mean Eoi$^{400}$-E$^{280}$ SST anomaly is 2.3°C with a global mean FCO$_2$ value of 0.56,
indicating that 56% of the warming (1.29°C) is predominantly driven by CO$_2$ forcing. Full interpretation of FCO$_2$ on SST is
presented in Burton et al. (2023).





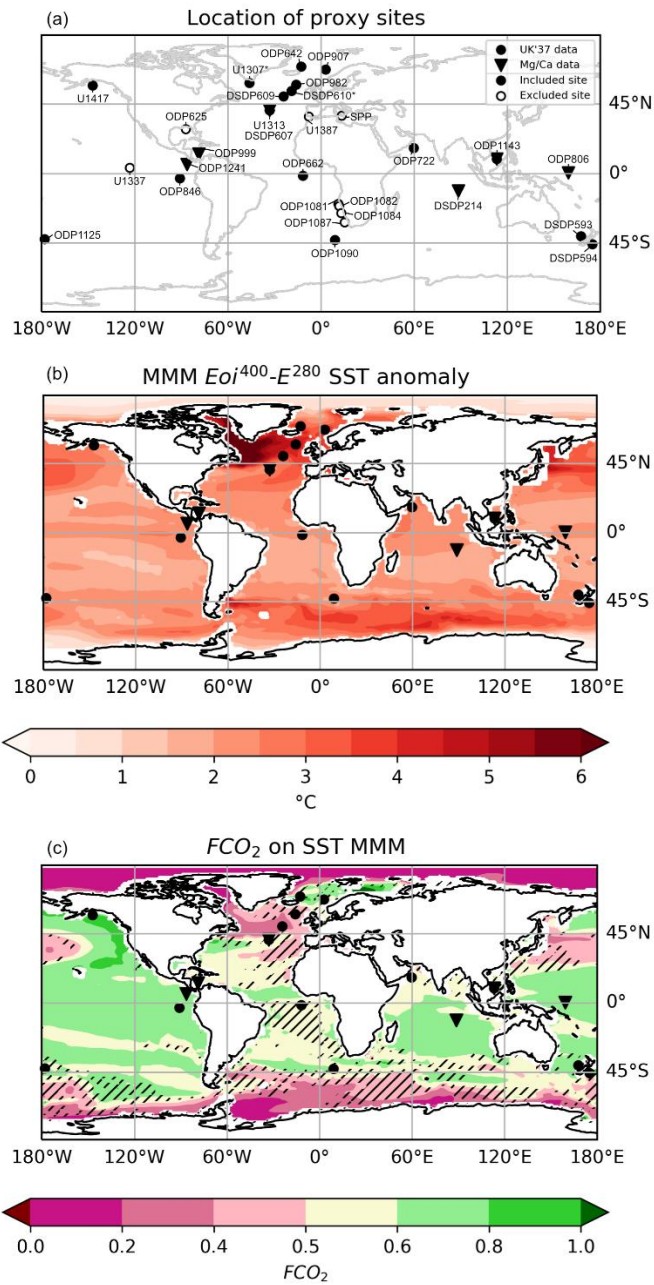

**Figure 1: Location of PlioVAR proxy data sites (a).** $U^{K'}_{37}$ **sites are denoted by a circle, and Mg/Ca sites are denoted a triangle. Filled symbols indicate that the site is used in this paper; open symbols indicate that the site is not used, either because no model SST values are available at the site and/or because analysis had not been conducted for the PRISM3 interval as well as KM5c. Sites marked with an asterisk (\*) only have data available for the PRISM3 interval (no data is available for KM5c) and are only considered in the temporal variability analysis (Section 3.3). Included sites are shown again in (b) with the MMM Eoi$^{400}$-E$^{280}$ SST anomaly and (c) with the FCO$_2$ on SST MMM. The MMM is comprised from CCSM4-UoT, CESM2, COSMOS, HadCM3, MIROC4m and NorESM1-F. Hatching in (c) denotes regions of uncertainty in FCO$_2$, defined where three or fewer models agreed on the dominant forcing (i.e., whether FCO$_2$ < 0.5 or FCO$_2$ > 0.5).**



$CO_2$ forcing is dominant in the low and mid-latitudes, and non-$CO_2$ forcing becomes more dominant in the high latitudes.

Given the spatial pattern of $FCO_2$ on SST (Fig. 1c), it is clear the lack of proxy data sites available in the high latitudes limits the identification of sites where SST is predominantly driven by non-$CO_2$ forcing. Aside from the North Atlantic, the other regions with low $FCO_2$ ($FCO_2 < 0.5$) – the Arctic Ocean, parts of the northern Pacific Ocean, and the Southern Ocean – have a relative dearth of proxy data sites available for the KM5c time slice.

The broad-scale pattern of $CO_2$ forcing being dominant at low and mid-latitudes and non-$CO_2$ forcing being dominant at

high latitudes persists throughout the year and does not change significantly between the seasons (Fig. 2). While the spatial patterns of $FCO_2$ on SST may not significantly change, the relative strength of the dominant forcing can be seen to differ.

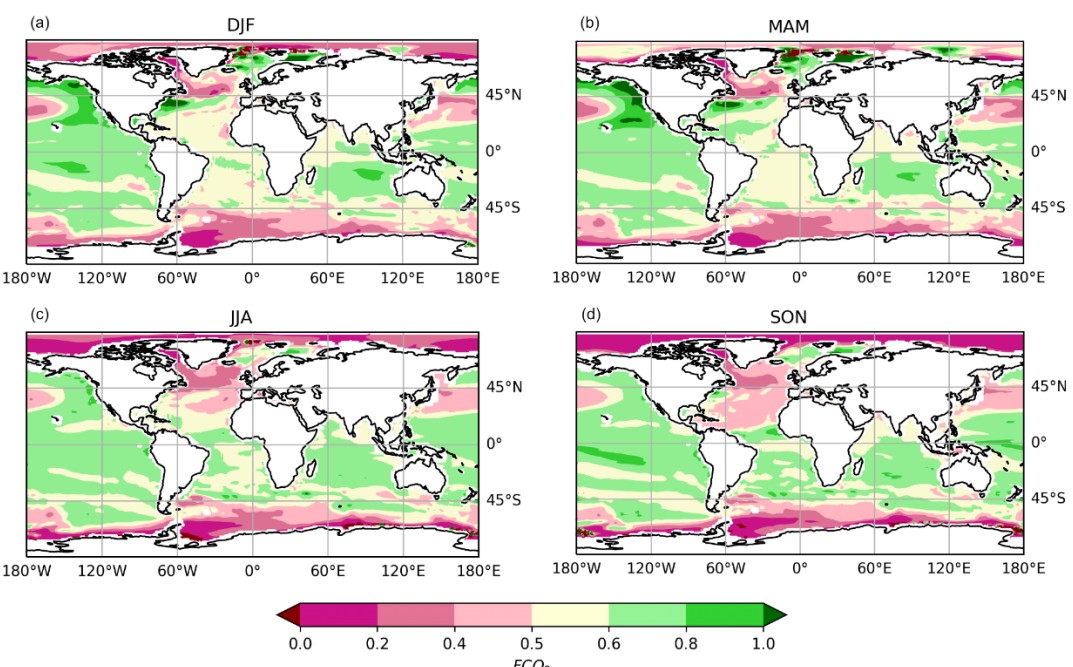

**Figure 2: Seasonal variation in $FCO_2$ on SST MMM shown for the months of December, January and February (DJF) (a), March,**
**April and May (MAM) (b), June, July and August (JJA) (c) and September, October and November (SON) (d). The MMM is comprised from CCSM4-UoT, CESM2, COSMOS, HadCM3, MIROC4m and NorESM1-F.**

In some regions, such as the North Atlantic, it is possible to see seasonal differences in both the spatial pattern of the dominant forcing and its relative influence. Non-$CO_2$ forcing is dominant in the northern North Atlantic basin in the months of December, January and February (DJF; Fig. 2a) and March, April and May (MAM; Fig. 2b), the influence of which

extends southward in June, July and August (JJA; Fig. 2c) to a maximum extent in September, October and November (SON; Fig. 2d). The region of low $FCO_2$ that extends throughout the year has relatively mixed forcing ($FCO_2$ 0.4-0.5), while a smaller region of more dominant non-$CO_2$ forcing ($FCO_2$ 0.2-0.4) remains relatively constrained between 45°N and 55°N, and 60°W to 20°W.




The FCO$_2$ method provides spatial detail in the drivers of climate change, but it alone cannot provide further information on
the specific mechanisms and processes behind the change(s) seen. The FCO$_2$ method allows us to comment on the collective
role of non-CO$_2$ forcing (representing both ice sheets and orography), but it does not allow us to comment on e.g., the role of
ice sheets alone, or the separate components encapsulated by 'orography' in the PlioMIP2 experiments (orography,
bathymetry, land-sea mask, lakes, soils, and prescribed vegetation; Haywood et al., 2016a). To do this, more model
experiments would be needed which further factorise the effects of ice vs. orography (see Haywood et al., 2016a).

### 3.1.1. FCO$_2$ on sea surface temperature at individual proxy data sites

The KM5c-PI SST anomaly at the majority of proxy data sites analysed (17 out of 19) is predominantly driven by CO$_2$
forcing (Table 3). Of these sites, one (U1417) was highly dominated by CO$_2$ forcing (FCO$_2$ 0.8-1.0), 10 were dominated by
CO$_2$ forcing (FCO$_2$ 0.6-0.8), and the remaining six sites experienced more mixed forcing (FCO$_2$ 0.5-0.6).

| FCO$_2$ | Interpretation | Sites | n sites |
|---|---|---|---|
| >1.0 | SST signal wholly dominated by CO$_2$ forcing with some non-CO$_2$ forcing acting in the opposite direction | - | 0 |
| 0.8-1.0 | SST signal highly dominated by CO$_2$ forcing (80-100% of signal caused by CO$_2$ forcing) | U1417 | 1 |
| 0.6-0.8 | SST signal dominated by CO$_2$ forcing (60-80% of signal caused by CO$_2$ forcing) | DSDP594*, DSDP593, ODP846, ODP806, ODP907*, DSDP214, ODP1241, ODP1143*, ODP1125, ODP1090 | 10 |
| 0.5-0.6 | Mixed forcing contributing to SST signal but CO$_2$ forcing dominant (50-60% of signal caused by CO$_2$ forcing) | ODP999, U1313*, DSDP607*, ODP722, ODP642, ODP662* | 6 |
| 0.4-0.5 | Mixed forcing contributing to SST signal but non-CO$_2$ forcing dominant (40-50% of signal caused by CO$_2$ forcing) | ODP982* | 1 |
| 0.2-0.4 | SST signal dominated by non-CO$_2$ forcing (20-40% of signal caused by CO$_2$ forcing) | DSDP609 | 1 |
| 0.2-0.0 | SST signal highly dominated by non-CO$_2$ forcing (0-20% of signal caused by CO$_2$ forcing) | - | 0 |
| <0.0 | SST signal wholly dominated by non-CO$_2$ forcing with some CO$_2$ forcing acting in the opposite direction | - | 0 |

Table 3: FCO$_2$ classes and their interpretation (adapted from Burton et al., 2023) with associated KM5c proxy data sites. Sites
marked with an asterisk (*) are in regions of uncertainty in FCO$_2$, defined where three or fewer models agreed on the dominant
forcing (i.e., whether FCO$_2$ < 0.5 or FCO$_2$ > 0.5).

The Southern Hemisphere mid-latitude sites (DSDP593, DSDP594, ODP1125 and ODP1090) are all dominated by CO$_2$
forcing (FCO$_2$ 0.6-0.8), as are the sites in the tropical Pacific (ODP806, ODP846, and ODP1241). Only sites ODP982 and
DSDP609 are predominantly influenced by non-CO$_2$ forcing, and both are situated in the North Atlantic. Further, the
majority of sites with FCO$_2$ between 0.5-0.6 are also interconnected via the low, mid and high latitude North Atlantic.





Each site was assessed for uncertainty in $FCO_2$ between the six models (i.e., whether $FCO_2 < 0.5$ or $FCO_2 > 0.5$ in three or fewer of the models (hatching in Fig. 1c)). Proxy sites are generally found where the models show agreement on the dominant forcing, however there are seven sites that do not. Of these seven sites, two have both $U_{37}^{K\prime}$ and Mg/Ca data available (U1313 and ODP1143) and five have only $U_{37}^{K\prime}$ data available (ODP907, ODP982, DSDP607, ODP662 and
DSDP594).

Of the 19 sites analysed, 11 had good data-model agreement between the reconstructed and simulated SST response (hereafter referred to as "data-model agreement"). Here we consider data-model agreement in terms of the difference between the MMM $Eoi^{400}$-$E^{280}$ SST anomaly and the KM5c proxy data-ERSSTv5 PI anomaly; sites that fall within ±2°C are considered to have good data-model agreement, and sites that fall within ±0.5°C are considered to have very good data-
model agreement (Table 4).

| Site | Lat. (°N) | Lon. (°E) | $FCO_2$ | Data-model agreement (°C) | |
|---|---|---|---|---|---|
| | | | | $U_{37}^{K\prime}$ | Mg/Ca |
| ODP907 | 69.24 | -12.70 | 0.66 | 2.08 | - |
| ODP642 | 67.22 | 2.93 | 0.56 | -2.65 | - |
| ODP982 | 57.52 | -15.87 | 0.44 | -1.37 | - |
| U1417 | 56.96 | -147.11 | 0.82 | 0.34 | - |
| DSDP609 | 49.88 | -24.24 | 0.27 | -0.08 | - |
| U1313 | 41.00 | -32.96 | 0.58 | -1.05 | -2.94 |
| DSDP607 | 41.00 | -32.96 | 0.58 | -0.53 | - |
| ODP722 | 16.60 | 59.80 | 0.58 | -0.36 | - |
| ODP999 | 12.74 | -78.74 | 0.58 | - | 5.34 |
| ODP1143 | 9.36 | 113.29 | 0.63 | -0.28 | 1.45 |
| ODP1241 | 5.84 | -86.44 | 0.63 | - | 3.19 |
| ODP806 | 0.32 | 159.36 | 0.71 | - | 2.59 |
| ODP662 | -1.39 | -11.74 | 0.55 | -0.64 | - |
| ODP846 | -3.09 | -90.82 | 0.71 | 0.66 | - |
| DSDP214 | -11.30 | 88.70 | 0.65 | - | 2.52 |
| DSDP593 | -40.51 | 167.67 | 0.76 | 2.43 | - |
| ODP1125 | -42.55 | -178.17 | 0.62 | -2.41 | - |
| ODP1090 | -42.91 | 8.90 | 0.60 | -1.63 | - |
| DSDP594 | -45.68 | 174.96 | 0.78 | 0.72 | - |

Table 4: Site-specific $FCO_2$ and data-model agreement at sites with KM5c data available.

Seven sites have good data-model agreement (light blue symbols in Fig. 3), and four have very good agreement (dark blue symbols in Fig. 3). The spatial distribution of these sites is representative of the total number of sites assessed, including the clustering of sites in the North Atlantic. In constraining the focus of our $FCO_2$ analysis to sites with good data-model
agreement (pie charts in Fig. 3) we should get the clearest and most accurate view of Late Pliocene SST change and its drivers.







**Figure 3: MMM Eoi$^{400}$-E$^{280}$ SST anomaly, represented by the background red shading. The MMM is comprised from CCSM4-UoT, CESM2, COSMOS, HadCM3, MIROC4m and NorESM1-F. Hatching represents uncertainty in FCO$_2$, where three or fewer of the six models agree on the dominant forcing (i.e., whether FCO$_2$ < 0.5 or FCO$_2$ > 0.5). The shape of the overlying symbols denotes the type of proxy data at each site (circle = $U^{K'}_{37}$, triangle = Mg/Ca); and the colour represents the level of data-model agreement (darker = stronger agreement). All proxy data is for KM5c.**

The FCO$_2$ on SST MMM is represented with a pie chart at each proxy site where there is good data-model agreement (i.e., the MMM Eoi$^{400}$-E$^{280}$ SST anomaly is within ±2°C of the proxy data SST anomaly). The proportion of the pie chart that is coloured denotes the proportion of total change attributable to CO$_2$ forcing (the FCO$_2$), also represented by the colour. Smaller pie charts with a dashed outline denote sites where there is uncertainty between models on the dominant forcing (i.e., where there is hatching on the main plot). Sites U1313 and ODP1143, marked by an asterisk (*), have both $U^{K'}_{37}$ and Mg/Ca data available; both the $U^{K'}_{37}$ data and the Mg/Ca data are within ±2°C for site ODP1143, but only the $U^{K'}_{37}$ data is within ±2°C for site U1313.

Sites ODP1143 and U1313 have both $U^{K'}_{37}$ and Mg/Ca data available. At site ODP1143, there is very good data-model agreement (within ±0.5°C) using the $U^{K'}_{37}$ data and good data-model agreement (within ±2°C) using the Mg/Ca data. There is





good data-model agreement (within $\pm 2°C$) using the $U^{K'}_{37}$ data at site U1313, but the Mg/Ca data has relatively poor data-model agreement ($>\pm 2°C$). The remaining sites with good data-model agreement are represented by $U^{K'}_{37}$ data only.

The range in $FCO_2$ on SST values indicate how the relative dominance of $CO_2$ forcing varies between all of the proxy sites, with $CO_2$ forcing driving between 27% and 82% of the total change seen. $CO_2$ is the dominant forcing accounting for the difference in SST between KM5c and PI at nine of the 11 sites with good data-model agreement (Fig. 3). Site U1417 is highly dominated by $CO_2$ forcing with a MMM $FCO_2$ on SST of 0.82; sites ODP1090, ODP846, ODP1143 and DSDP594 are dominated by $CO_2$ forcing; and sites ODP662, DSDP607, U1313 and ODP722 have more mixed forcing, though $CO_2$

remains dominant. In contrast, sites DSDP609 and ODP982 in the North Atlantic are predominantly driven by non-$CO_2$ forcing, the only two proxy sites included in this study where this is the case.

It is worth noting that six of these sites (ODP662, DSDP607, U1313, ODP982, ODP1143 and DSDP594) show uncertainty in the $FCO_2$ on SST between models (smaller pie charts with dashed outlines in Fig. 3; see Table S2 and Fig. S2 in the Supplement for individual model values). This means that three or fewer of the six models agree on the dominant forcing

(i.e., whether $FCO_2 > 0.5$ or $FCO_2 < 0.5$) and hence conclusions should be drawn with caution given the uncertainty between the models.

### 3.2. Proxy data-model agreement and $FCO_2$ on sea surface temperature

All KM5c proxy data sites were explored to assess whether the $FCO_2$ method could provide insight into the reason for the (lack of) data-model agreement. For example, if the data-model agreement is better where $FCO_2$ is high, then the non-$CO_2$

forcing in the models may be inaccurate. Given the differences in oceanographic settings of the proxy data sites included, we hypothesise that any relationship would be site dependent.

To assess whether there was a relationship between $FCO_2$ on SST and data-model agreement, the $FCO_2$ on SST and $Eoi^{400}$-$E^{280}$ anomaly for each of the six models were also individually assessed. Whether data-model agreement was very good, good, or poor there was no consistent or significant relationship between $FCO_2$ and data-model agreement (Fig. 4). Values of

$FCO_2$ above 1 and below 0 had the potential to skew any relationships and were seen at multiple sites and in multiple models.





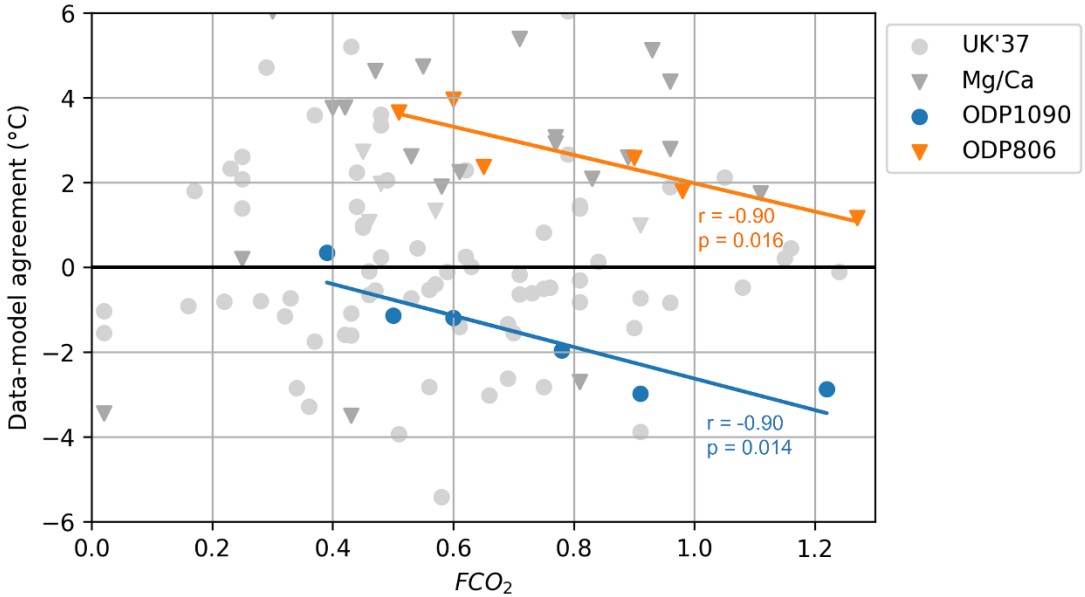

**Figure 4: The relationship between individual model FCO₂ and the model-proxy data anomaly for all KM5c proxy sites considered here. Only the two sites with a significant relationship are coloured:** $U^{K'}_{37}$ **site ODP1090 in blue circles and Mg/Ca site ODP806 in**

**orange triangles. Note that the FCO₂ scale extends to 1.3 to include values from all six models. Some data points with extreme FCO₂ values lie outside of the FCO₂ range included in this figure, but the axes limits allow the significant relationships at sites ODP1090 and ODP806 to be demonstrated clearly.**

Only two sites showed a statistically significant relationship: ODP1090 (blue circles in Fig. 4) and ODP806 (orange triangles in Fig. 4). There is a negative relationship between FCO₂ and data-model agreement at site ODP1090 (r = -0.90, p = 0.014;

blue in Fig. 4). Data-model agreement varies from 0.35°C (CESM2) to -2.98°C (NorESM1-F) with a MMM of -1.63°C, and FCO₂ on SST varies from 0.39 (CESM2) to 1.22 (COSMOS) with a MMM of 0.60. If COSMOS is excluded due to the FCO₂ value above 1, the relationship further strengthens (r = -0.96, p = 0.0023). CESM2, the only model with an FCO₂ value less than 0.5 (indicating that non-CO₂ forcing is dominant), has the best data-model agreement. This relationship – of models with higher FCO₂ having worse data-model agreement – may suggest that the influence of CO₂ forcing is overestimated in

some of the models and/or that the models underestimate the influence of non-CO₂ forcings at site ODP1090.

There is also a negative relationship between FCO₂ and data-model agreement at site ODP806 (r = -0.90, p = 0.016; orange in Fig. 4). Data-model agreement varies from 1.17°C (NorESM1-F) to 3.96°C (COSMOS) with a MMM of 2.59°C, and FCO₂ on SST varies from 0.51 (CESM2) to 1.27 (NorESM1-F) with a MMM of 0.71. NorESM1-F has the best data-model agreement and the highest FCO₂ value; even if it is excluded because of the FCO₂ value above 1, the relationship remains

strong and statistically significant (r = -0.81, p = 0.048). This relationship – of models with higher FCO₂ having better data-model agreement – may suggest that non-CO₂ forcing has too large an impact at site ODP806 in models with lower FCO₂. It is worth noting, however, that site ODP806 is the only site assessed where all six models agree that CO₂ is the dominant forcing, i.e., where all six models have an FCO₂ on SST > 0.5 (CESM2 has the lowest FCO₂ on SST value at 0.51).



Though the subset of models considered here is shown to be representative of the whole PlioMIP2 ensemble (Table 2), our
confidence in the relationship between $FCO_2$ and data-model agreement seen at sites ODP1090 and ODP806 is linked to the
overall sample size and inherent uncertainty in both the models and proxy data. The hypothesis that the relationship would
be site dependent is found to be true, though many of the relationships seen were not statistically significant so our
confidence in this conclusion is limited. It also appears that the relationship may be dependent on the proxy type: the cool
bias in the Mg/Ca SST data discussed in McClymont et al. (2020) is visible in Fig. 4 (Mg/Ca data represented by the grey
triangles), with the majority of data-model comparison values above 0°C. Mg/Ca data also appear to have poorer data-model
agreement than the $U_{37}^{K\prime}$ data, though commenting on the reasons for this is beyond the scope of this paper.

### 3.3. Temporal variability

We hypothesise that sites with a lower $FCO_2$ (i.e., sites where non-$CO_2$ forcing was more dominant) could experience greater
temporal variability in forcing, and therefore in temperature response to forcing feedbacks. This is because, on orbital
timescales, there could be changes in the ice-sheet and vegetation components of the non-$CO_2$ forcing, and/or changes in sea
ice. Changes in ice sheets, vegetation and sea ice are more likely to affect the regions that are more influenced by non-$CO_2$
forcing (i.e., regions with lower $FCO_2$). These are mainly the higher latitudes, but more distant regions can also be affected
by the movement of the thermal equator that occurs with polar ice changes.

By including and comparing SST proxy data from the PRISM3 interval (3.264-3.025 Ma), it is possible to comment on the
temporal variability seen in the SST proxy data results and how that compares and relates to the $FCO_2$ analysis. This was
investigated by using standard deviation as an approximation for temporal variability (Fig. 5).





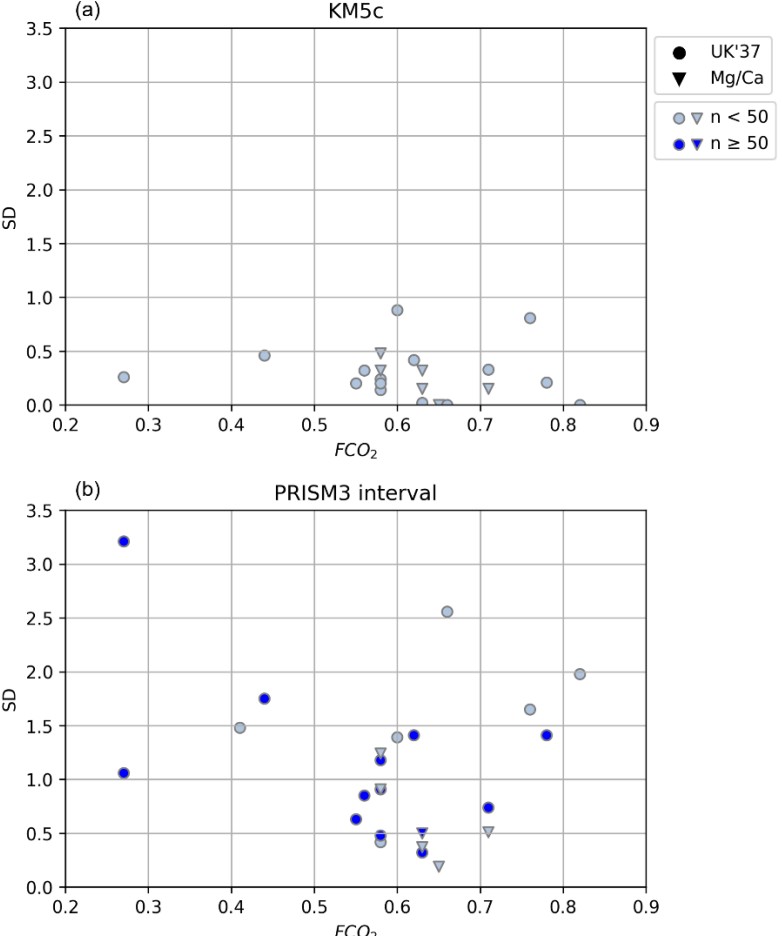

**Figure 5: FCO₂ on SST MMM compared to standard deviation (SD) of SST proxy data for KM5c (a) and the PRISM3 interval (b).**
**$U_{37}^{K'}$ data is represented by circles and Mg/Ca data is represented by triangles. Sites are grouped into two classes according to the**
**number of data points available: sites with fewer than 50 data points are shown in light blue, and sites with greater than or equal**
**to 50 data points are shown in dark blue.**

The relationship between FCO₂ and standard deviation is highly sensitive to the sample size, and it is clear that an increase in

both the number of sites and the number of data points at each site is needed to explore this relationship further. If one

focuses on KM5c, the maximum sampling density at a given site is 10 ($U_{37}^{K'}$ at sites ODP1125 and ODP722). It was therefore

necessary to consider the PRISM3 interval to capture a greater range in standard deviation, and hence temporal variability

estimates. Furthermore, because KM5c was chosen as a target for analysis in part due to its low variability (Haywood et al.,

2013; McClymont et al., 2020), one would only expect significant variability to be seen in the PRISM3 interval. 21 proxy

data sites have data available for the PRISM3 interval, with sampling densities ranging between 4 (Mg/Ca at site DSDP214)

and 125 ($U_{37}^{K'}$ at site ODP722).





In total – accounting for both $U_{37}^{K'}$ and Mg/Ca data and the two time intervals – 15 sites have a sample size less than or equal

to 5, and 12 sites have a sample size between 5 and 25 (Table 5; see Table S3 in the Supplement for site names). If we

exclude sites with a sample size of less than 50 (by default therefore looking only at the PRISM3 interval), it is possible to

see the hypothesised relationship of sites with a higher $FCO_2$ having a lower temporal variability. For the 13 sites where the

sample size is greater than or equal to 50 (dark blue symbols in Fig. 5), the hypothesised relationship is seen (r = -0.56, p =

0.070). Though the hypothesised relationship – low $FCO_2$ sites experience greater temporal variability – is seen our

confidence is limited due to the relatively low data availability, highlighting the need for more data availability at both

existing and new proxy sites. No relationship is seen for sites with a sample size of less than 50, whether data from KM5c is

included (r = 0.10, p = 0.59) or excluded (r = 0.18, p = 0.32).

|  | Number of $U_{37}^{K'}$ sites | | Number of Mg/Ca sites | |
| --- | --- | --- | --- | --- |
| n | KM5c | PRISM3 | KM5c | PRISM3 |
| n ≤ 5 | 8 | 1 | 5 | 1 |
| 5 < n ≤ 25 | 7 | 3 | 1 | 1 |
| 25 < n ≤ 50 | 0 | 2 | 0 | 3 |
| 50 < n ≤ 100 | 0 | 10 | 0 | 1 |
| n > 100 | 0 | 2 | 0 | 0 |

Table 5: Sampling densities at proxy sites. Note that two sites (U1313 and ODP1143) have $U_{37}^{K'}$ and Mg/Ca data available for both
KM5c and the PRISM3 interval, and a further site (ODP999) has only Mg/Ca data available for KM5c but both Mg/Ca and $U_{37}^{K'}$
data available for the PRISM3 interval.

More work is needed to further explore this relationship. In particular, future modelling efforts could calculate $FCO_2$ from

other time slices within the PRISM3 interval, as the $FCO_2$ results here only represent the influence of forcing on the climate

during KM5c. These results are presented here with an awareness of this limitation, and nonetheless still represent the best

exploration possible given the model and proxy data currently available.

## 4. Discussion

### 4.1. FCO₂ and seasonality

We show that $CO_2$ forcing is the main driver of SST change for MIS KM5c relative to the PI at most of the proxy sites

assessed (Section 3.1.1), and that $FCO_2$ on SST varies seasonally at a global scale (Section 3.1). However, to further explore

the potential for seasonal reconstructions of $FCO_2$ to inform the interpretation of specific SST proxy records, it is necessary

to consider $FCO_2$ at a local (site-specific) level. We present seasonality in both $FCO_2$ on SST MMM and the MMM Eoi[400]-

E[280] SST anomaly for three individual proxy sites which provides a possible framework for the discussion regarding the

climate signal recorded in the proxy data. Though the seasonality in SST is model-dependent, the models are consistent

enough to begin to suggest trends that may be useful when interpreting the proxy data.

Three $U_{37}^{K'}$ sites are selected as examples (Fig. 6), though it is possible to conduct this analysis for any of the sites included in

this paper. Site DSDP609, in the North Atlantic, has the lowest annual mean $FCO_2$ on SST (0.27) in the collection of sites





presented. Site U1417, in the Gulf of Alaska, has the highest annual mean FCO$_2$ on SST (0.82). Site ODP1090 is one of the most southern sites presented and has an annual mean FCO$_2$ on SST of 0.60. The proxy data is taken to reflect the annual mean and proxy values of each of these sites is presented with two uncertainty estimates, ±1°C and ±2°C (orange shading in

Fig. 6). Calculating site- and proxy-data-type specific uncertainties is beyond the scope of this paper, so these uncertainty estimates represent a simple thought experiment that approximate plausible uncertainty values purely to provide the necessary context (i.e., magnitude of seasonality seen in FCO$_2$ vs. limitations of reproducing the magnitude of temperature changes from the proxy record).





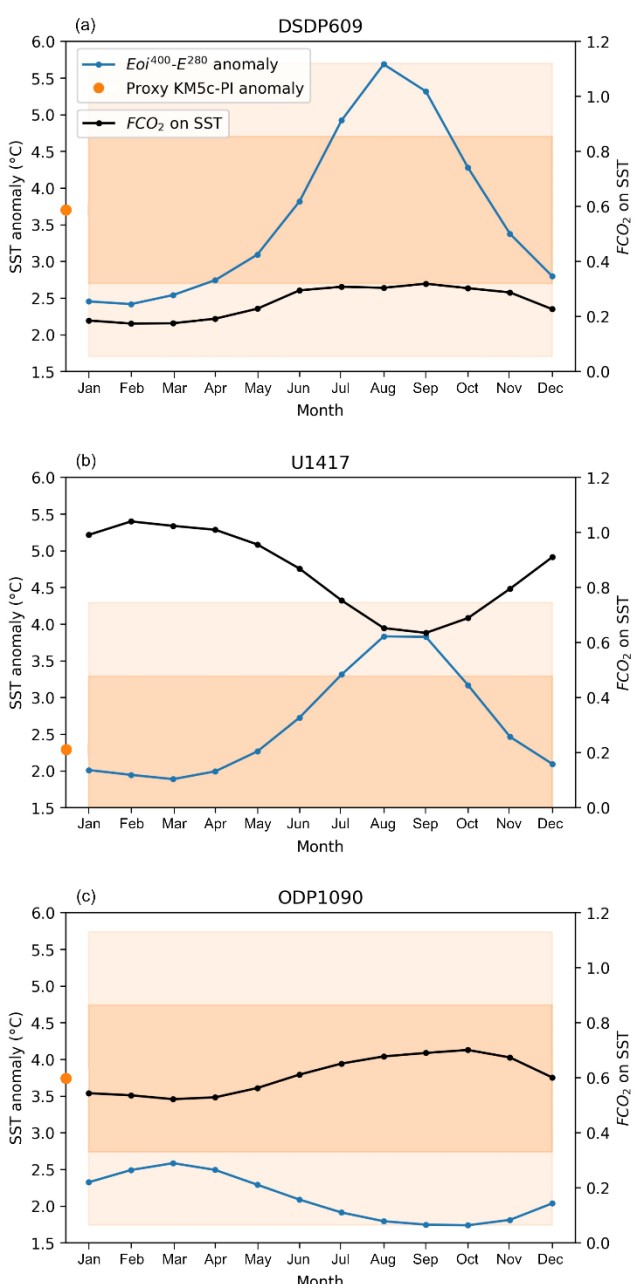

**Figure 6: Monthly Eoi$^{400}$-E$^{280}$ SST anomaly (blue line) and FCO$_2$ on SST (black line) at sites DSDP609 (a), U1417 (b) and ODP1090 (c). The proxy data KM5c-PI anomaly value is shown by the orange circle on the y-axis, with plausible uncertainty estimates shown by orange shading (darker orange denoting ±1°C, lighter orange denoting ±2°C). Note that the scale for FCO$_2$ on SST extends to 1.2 as some monthly values exceed 1.0 for site U1417.**

There is a large seasonal variation in the Eoi$^{400}$-E$^{280}$ anomaly at site DSDP609 (Fig. 6a), with the smallest anomaly (2.45°C) in January and maximum anomaly (5.69°C) in August. The proxy data anomaly of 3.70°C is well matched to the MMM





annual mean Eoi$^{400}$-E$^{280}$ anomaly of 3.62°C, and all the model seasonality is captured within ±2°C of the proxy data value (blue line within orange shading in Fig 6a). Though there is large seasonal variation in the Eoi$^{400}$-E$^{280}$ anomaly, the total range in FCO$_2$ on SST is relatively small. FCO$_2$ increases between January (0.18) and September (0.32), suggesting that CO$_2$ forcing has a proportionally greater role in the warming in this period. Despite this, it is important to note that site DSDP609

is always either highly dominated (FCO$_2$ 0.0-0.2) or dominated (FCO$_2$ 0.2-0.4) by non-CO$_2$ forcing.

The magnitude of the seasonal variation in the Eoi$^{400}$-E$^{280}$ anomaly is smaller at site U1417 than at site DSDP609, but there is greater variation in FCO$_2$ on SST (from 0.63 (September) to 1.04 (February), Fig. 6b). The smallest Eoi$^{400}$-E$^{280}$ anomalies are seen at the start of the year and culminate in the greatest warming in the late summer and early autumn months. The proxy data anomaly of 2.29°C is smaller than the mean annual MMM anomaly of 2.63°C, which could suggest that the

models overestimate the magnitude of the summer/autumn peak in warming and/or that the peak is not fully represented in the proxy data, but the data-model agreement is very good and all model seasonality is within ±2°C of the proxy data (blue line within orange shading in Fig. 6b). FCO$_2$ clearly decreases throughout MAM and JJA as the Eoi$^{400}$-E$^{280}$ anomaly increases, indicating that the greatest warming is attributable to changes in non-CO$_2$ forcing. Interestingly, FCO$_2$ on SST is above 1 for February (1.04), March (1.02) and April (1.01), indicating that there is a small role of non-CO$_2$ forcing in

cooling the SST given the overall warming signal. However, the overall signal of change is dominated by CO$_2$ forcing throughout the year and the lowest FCO$_2$ on SST is 0.55 in May (indicating mixed forcing with CO$_2$ forcing dominant).

Compared to DSDP609 and U1417 in the Northern Hemisphere mid latitudes, site ODP1090 shows little seasonal variation in the Eoi$^{400}$-E$^{280}$ anomaly (Fig. 6c). The greatest warming is seen in the summer (DJF) and autumn (MAM) seasons, with a total variation from 1.74°C in October to 2.58°C in March. The proxy data anomaly of 3.74°C is over a degree warmer than

the warmest month in the model data (March, 2.58°C) which may indicate either that the models do not accurately represent the degree of warming and/or that the proxies overestimate the warming. The months with least warming in the models (August-November) are at the low end of the plausible proxy data uncertainties used (blue line within light orange shading in Fig. 6c). The FCO$_2$ remains fairly consistent throughout the year (ranging from 0.52 (March) to 0.70 (October)) and CO$_2$ is always the dominant forcing. Months with the lowest FCO$_2$ are also the months with the greatest Eoi$^{400}$-E$^{280}$ anomaly,

indicating that this warming can be attributed non-CO$_2$ forcing (e.g., changes to positions of the front systems associated with the Antarctic circumpolar current or Antarctic ice sheet, though the FCO$_2$ method alone cannot detail the exact non-CO$_2$ forcing components).

These results highlight the usefulness of the FCO$_2$ method in terms of understanding the drivers of seasonal trends in SST change at the individual site level. Though it is beyond the scope of this paper, comparing the model-derived SST anomaly

and FCO$_2$ with different proxy systems and/or different calibration methods used within a particular proxy system may shed light on and resolve proxy data biases, as well as data-model discrepancies. Furthermore, it may be interesting to complete a seasonal analysis for sites with different oceanographic settings; the three sites presented here represent a range in FCO$_2$ but may not be representative of the range in seasonality seen at all proxy sites.





### 4.2. Constraints on site-specific climate sensitivity estimates

There is a discernible relationship between ECS and the ensemble-simulated Pliocene surface air temperature (SAT) anomaly within the PlioMIP2 ensemble (Haywood et al., 2020). Quantifying the influence of $CO_2$ forcing at individual proxy data sites using the $FCO_2$ method means that it is now possible to better determine which sites are potentially best placed to inform estimates of ECS.

The modelled tropical oceans are particularly strongly related to modelled ECS, indicating that it is possible to constrain
estimates of ECS using Pliocene SST data from the tropics. Haywood et al. (2020) used the Foley and Dowsett (2019) SST reconstruction (hereafter "FD19") in this way. They adapted the methodology of Hargreaves and Annan (2016), whereby

$$ECS = \alpha\Delta T(30°N - 30°S) + C + \varepsilon, \tag{1}$$

where $\alpha$ and $C$ are constants and $\varepsilon$ represents all errors in the regression equation. This methodology was adapted to account for the more sparsely distributed proxy data in PlioMIP2 compared to PlioMIP1 (due to the change in target from the
PRISM3 time slab (3.264-3.025 Ma) to the KM5c time slice (3.205 Ma ± 0.01 Ma)), and instead relies on point-based observations and local regressions between $Eoi^{400}$-$E^{280}$ SST and modelled ECS.

The adapted methodology applies Equation 1 with ΔSST from individual data sites, and $\alpha$ and $C$ are location dependent, meaning that sites northward of 30°N and southward of 30°S can also be considered. This produces a different estimate of ECS for each proxy site, though this does not imply that ECS is different for each location (Haywood et al., 2020). Data is
presented for sites which meet two conditions: 1) that the relationship between the site-specific $Eoi^{400}$-$E^{280}$ SST anomaly and a model's ECS was significant at the 95% confidence interval, and 2) that at least one of the models in the PlioMIP2 ensemble was within ±1°C of the proxy data (Haywood et al., 2020). If a proxy site fell on land in the Pliocene land-sea mask in the models, the nearest ocean grid point value was taken.

Here we repeat the Haywood et al. (2020) methodology using the PlioVAR data used throughout this paper and the full suite
of models in the PlioMIP2 ensemble (see Haywood et al. (2020) for details of the ensemble). We assess the PlioVAR data ECS estimates in the context of $FCO_2$ on SST (Fig. 7a), and compare these results to the original ECS estimates generated from FD19 presented in Haywood et al. (2020) (Fig. 7b).

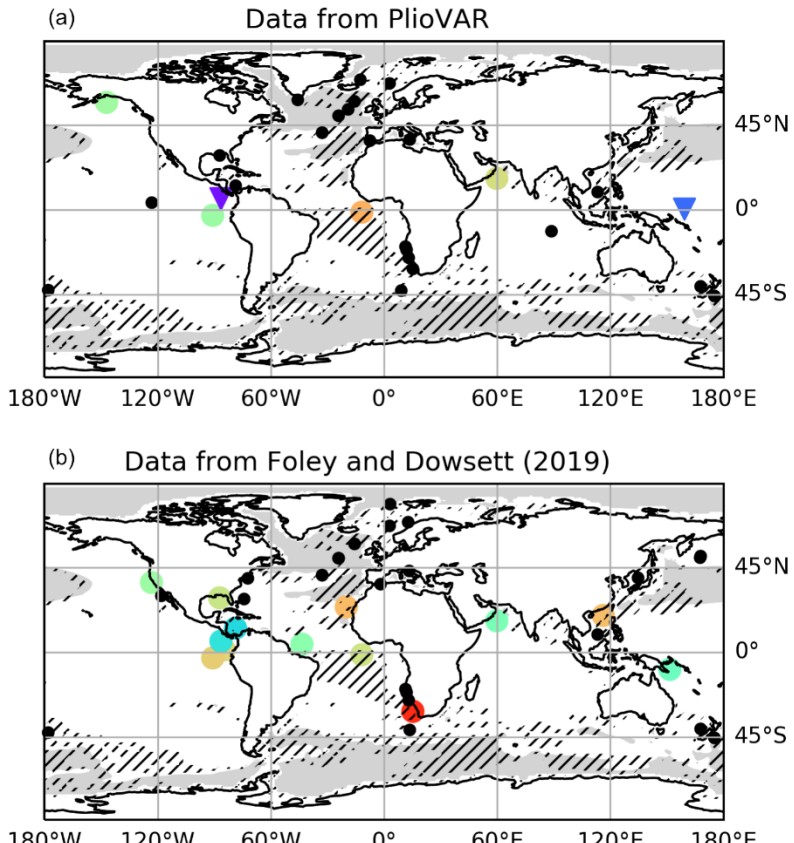

**Figure 7: ECS estimates using the BAYSPLINE reconstruction in McClymont et al. (2020) and Mg/Ca reconstruction in McClymont, Ho et al. (2023) (a), and ECS estimates using FD19 presented in Haywood et al. (2020) (b). The triangles in (a) represent Mg/Ca sites, all other sites have $U_{37}^{K'}$ data. The smaller black circles in represent sites that did not meet the conditions so were excluded. FCO₂ on SST is shown by the background shading: areas in white are predominantly driven by CO₂ (FCO₂ > 0.5) and areas in light grey are predominantly driven by non-CO₂ forcing (FCO₂ < 0.5). Hatching represents where three or fewer models agree on the dominant forcing (i.e., whether FCO₂ < 0.5 or FCO₂ > 0.5).**

Different proxy sites are represented in the results depending on the dataset chosen given that the proxy data had to be within ±1°C of the Eoi[400]-E[280] anomaly for at least one model to be included (i.e., within ±1°C of the model that shows the most or least change). It is important to note that this methodology compares the proxy data to the Eoi[400]-E[280] anomalies for all 17 of the models in the PlioMIP2 ensemble rather than the subset of six models used in the remainder of this paper, hence some sites are included in the climate sensitivity analysis that are not included elsewhere to maximise the number of sites available.





Thirteen sites are represented using FD19 (Fig. 7b), compared to six sites using the PlioVAR data (Fig. 7a). This reduced
number partly reflects the additional age constraints applied to the PlioVAR data (see McClymont et al., 2020). Four sites
are represented in both datasets: ODP1241, ODP662, ODP846, and ODP722. Five FD19 sites (ODP625, ODP1018,
ODP1087, ODP1115 and ODP1146) met the criteria but the Pliocene land-sea mask in the models meant that FCO$_2$ on SST
was not available; as Burton et al. (2023) show that FCO$_2$ on SAT is comparable to FCO$_2$ on SST outside of the high
latitudes, the FCO$_2$ on SAT was taken at these sites as the closest approximation for the climate sensitivity analysis only.

All FD19 data is $U_{37}^{K\prime}$ using the Müller et al. (1998) calibration, and of the six sites using PlioVAR data, four have $U_{37}^{K\prime}$ data
(BAYSPLINE calibration) and two have Mg/Ca data. Though the $U_{37}^{K\prime}$ datasets use different calibrations, the difference in
reconstructed temperatures is small (see Supplement of McClymont et al., 2020). The two Mg/Ca sites do not meet the exact
conditions used by Haywood et al. (2020) but are included to begin to explore the applicability of this method to other proxy
types; rather than being within ±1°C of one of the models, the SST proxy data was 1.12°C and 1.20°C cooler than the MMM
Eoi$^{400}$-E$^{280}$ SST anomaly at sites ODP806 and ODP1241, respectively.

Individual site ECS estimates are shown in Table 6. The original range in estimates presented in Haywood et al. (2020) using
FD19 is 2.63°C to 4.80°C. The range broadens to 1.59°C to 4.15°C using the PlioVAR data, but this is skewed by the two
Mg/Ca sites with ECS estimates of 1.59°C and 1.99°C at sites ODP1241 and ODP806, respectively. Both of these ranges are
broader but generally align with the *likely* range of ECS presented in the Sixth Assessment Report of the Intergovernmental
Panel on Climate Change (IPCC) of 2.5°C to 4.0°C (Arias et al., 2021). If only the $U_{37}^{K\prime}$ PlioVAR data is considered here
(i.e., only those data that meet the original condition of being within ±1°C of one of the models), the range in ECS estimates
is constrained to 3.44°C to 4.15°C, one of the best constrained estimates of Pliocene ECS to date.

| Site | Lat (°N) | Lon (°E) | ECS estimate (°C) | |
| --- | --- | --- | --- | --- |
| | | | FD19 | PlioVAR |
| U1417 | 56.96 | -147.11 | - | 3.47 |
| ODP1018* | 36.99 | -123.28 | 3.45 | - |
| ODP625* | 28.83 | -87.16 | 3.75 | - |
| ODP958 | 24.00 | -20.00 | 4.08 | - |
| ODP1146* | 19.46 | 116.27 | 4.07 | - |
| ODP722 | 16.62 | 59.80 | 3.22 | 3.83 |
| ODP999 | 12.75 | -78.73 | 2.63 | - |
| ODP1241^ | 5.85 | -86.45 | 2.72 | 1.59 |
| ODP925 | 4.20 | -43.49 | 3.33 | - |
| ODP677 | 1.20 | -84.73 | 3.9 | - |
| ODP806^ | 0.32 | 159.36 | - | 1.99 |
| ODP662 | -1.39 | -11.74 | 3.78 | 4.15 |
| ODP846 | -3.09 | -90.82 | 3.96 | 3.44 |
| ODP1115* | -9.19 | 151.57 | 3.12 | - |
| ODP1087* | -31.47 | 15.32 | 4.80 | - |

**Table 6: Proxy sites and their ECS estimates using FD19 and/or the PlioVAR data used in this paper. Sites marked with an asterisk (*) are on land in the model Pliocene land-sea mask. Mg/Ca sites are marked with a caret (^).**





Regardless of the estimate of ECS itself, all of the sites selected using the adapted methodology are in regions where $CO_2$ is
the dominant forcing (i.e., where $FCO_2 > 0.5$; ocean regions in white in Fig. 7). Site ODP662 (present in both the PlioVAR
dataset and FD19) has the lowest $FCO_2$ on SST at 0.55, while sites U1417 and ODP846 have the highest $FCO_2$ on SST in the
PlioVAR dataset and FD19, respectively, with values of 0.82 and 0.71. Uncertainty in $FCO_2$ on SST (hatching in Fig. 7) is
only seen at site ODP662; the remaining sites selected using the adapted methodology show consistent agreement on the
dominant forcing of site-specific SST change in at least four of the models.

Sites excluded for not meeting the conditions (small black circles in Fig. 7) are generally found in regions of low $FCO_2$
($FCO_2 < 0.5$; white shading in Fig. 7) and/or in regions of uncertainty in $FCO_2$ (hatching in Fig. 7). Site DSDP214 in the
Indian Ocean is an example of an exception to this general rule but, despite the $FCO_2$ on SST being 0.65 indicating that $CO_2$
forcing is dominant, it is likely to also be influenced by gateway changes (e.g., Karas et al., 2009, 2011). We find that the
Mg/Ca data at site DSDP214 does not provide good data-model agreement (2.52°C; Table 4), and Mg/Ca data sites also
appear to generate notably different ECS estimates than $U^{K'}_{37}$ sites.

In this way, the $FCO_2$ method could potentially support the selection of sites used in the Haywood et al. (2020) methodology
for constraining estimates of ECS. At present there is not enough data available to explore whether the proxy-informed
estimate of ECS is related to $FCO_2$ but given that there is no significant relationship between ECS and $FCO_2$ on SST in the
models used here (see Burton et al., 2023) the potential for finding such a relationship may be unlikely.

**5. Summary and future work**

We have assessed the role of $CO_2$ forcing in SST patterns in the Pliocene by using the recently introduced $FCO_2$ method
(Burton et al., 2023) and the current best available proxy data (McClymont et al., 2020; McClymont, Ho et al., 2023). We
focused on SST patterns due to the relative wealth of KM5c-age proxy data, but a similar exploratory approach could also be
adopted for other proxies that provide a quantitative measure of a climate variable, e.g., for SAT or precipitation.

By using the climate models and proxy data in tandem, we have explored the forcings behind the SST patterns more than has
previously been possible. We show that the majority of proxy sites are predominantly forced by $CO_2$, and those sites that are
not are only found in the North Atlantic. We have also presented site-specific seasonal analysis using model data which
provides a potential way to gain further insight into how changes in seasonality could be reflected in the reconstructed
annual mean temperature change. Both of these components have allowed us to highlight potential reasons for data-model
agreement (or lack thereof). A well-constrained, proxy-informed estimate of Pliocene ECS is also presented. Using the latest
PlioVAR data, ECS is estimated to be within the range of 1.59°C to 4.73°C, constrained to 3.44°C to 4.73°C if only $U^{K'}_{37}$ data
is considered (as in the original methodology presented in Haywood et al. (2020)).

It is recommended that future work builds on the foundation presented here: the conclusions drawn would be strengthened if
more data was available, both from more proxy data sites and from more models running the necessary experiments to apply



the FCO$_2$ method. This would set both the palaeoclimate modelling and proxy data communities on track to best understand the climate of the Pliocene, and to improve data-model comparison efforts.

**Data availability**

The model data required to produce the FCO$_2$ results in this paper is available in the Supplement. The proxy data from McClymont et al. (2020) is available at https://doi.pangaea.de/10.1594/PANGAEA.911847. The proxy data from
McClymont, Ho et al. (2023) is available at https://doi.pangaea.de/10.1594/PANGAEA.956158. The proxy data from Foley and Dowsett (2019) is available at https://www.sciencebase.gov/catalog/item/5d0a7ac8e4b0e3d3115ff62b.

**Author contribution**

LEB led the study, analysed the data and contributed to the writing of the manuscript. AMH, JCT, AMD and DJH helped to prepare the paper with contributions from all co-authors. ELM, SLH and HLF supported the use and integration of marine
proxy sea surface temperature data into the manuscript and contributed to the description of the data as outlined within the paper.

**Competing interests**

At least one of the (co-)authors is a member of the editorial board of *Climate of the Past*.

**Acknowledgements**

For the purpose of open access, the author has applied a Creative Commons Attribution (CC BY) licence to any Author Accepted Manuscript version arising. Lauren E. Burton acknowledges that this work was supported by the Leeds-York-Hull Natural Environment Research Council (NERC) Doctoral Training Partnership (DTP) Panorama under grant NE/S007458/1. Heather L. Ford acknowledges the Natural Environment Research Council under grant NE/N015045/1. The collation and
analysis of the PlioVAR data is an outcome of the working group Pliocene Climate Variability over glacial-interglacial timescales, sponsored by Past Global Change (PAGES). We acknowledge PAGES for their financial support to workshops and discussions, and thank all of the PlioVAR members who created, synthesised, reviewed and analysed the proxy data to generate the PlioVAR data. This research used samples and/or data provided by the International Ocean Discovery Program (IODP), Ocean Drilling Program (ODP), and Deep Sea Drilling Project (DSDP). We acknowledge the PlioMIP2 participants
whose data has been used in the support of this study.





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
