# Peer review of "The role of atmospheric CO2 in controlling sea surface temperature change during the Pliocene"

_Climate of the Past, 2023_

## Author Response (AR1)

**The role of atmospheric $CO_2$ in controlling sea surface temperature change during the Pliocene**

**Revision**

**General**

Line 11 – Correction to the number of models considered in this paper.

Figure 3 has been corrected to show $FCO_2$ on SST values (rather than percentages) in the pie charts.

General corrections for improved clarity throughout the manuscript.

**Reviewer 1**

The authors thank Anonymous Referee #1 for their comments and feedback. Revisions are described below and indicated in the revised manuscript.

Additional detail and references to Budyko (1982) and Zubakov and Borzenkova (1988) have been added in Line 26/27. Full consideration of the analogue concept is beyond the scope of this paper, but work is in preparation that addresses this important point.

The citation in Line 35, and associated entry in the reference list, has been updated from Haywood et al. (in press) to Haywood et al. (2024).

The suggested sentence starting on Line 54 has been removed.

The isotope example has been reworded for improved clarity in Line 90/91.

The citation in Line 183/184 has been corrected to Dowsett et al. (2010).

**Reviewer 2**

The authors thank Anonymous Referee #2 for their comments and feedback. Revisions are described below and indicated in the revised manuscript.

Additional detail has been added to Section 1 which discusses the pattern of SST change in the Pliocene, and some fundamental questions of Pliocene climate change.

The title of the paper and references to "patterns" of SSTs have been rephrased to avoid confusion with the SST pattern effect, which is not centrally examined in this paper. "Patterns" was originally used to describe the global-scale picture of SSTs. We have now included comments on some zonal and meridional gradients seen in the Pliocene, but the focus of the paper remains on site-specific SSTs and the overall global picture. Future research could use the $FCO_2$ analysis to consider the drivers of zonal and meridional changes in more detail.

Lines 85-100 detail examples of previous data-model comparison efforts and hence provide important context for the novel data-model comparison analysis presented in this paper.

**Tim Herbert**

Many thanks for your comment and feedback, Tim.

For reproducibility we decided to only include sites in the published KM5c alkenone database in McClymont et al. (2020), though future research could apply this method to any site for which there is the necessary data.

On the North Atlantic, we do see variation between the models used in this study in terms of the $FCO_2$ on SST as well as the Pliocene minus pre-industrial SST anomaly. It is not yet possible for us to run the resolution of model required to accurately capture the North Atlantic Current to equilibrium, so the results we have are the best approximation available given current capability. This will undoubtedly have an impact on the results, but by considering an ensemble of models (rather than an individual model) we are more aware of the differences and subsequent uncertainty. The $FCO_2$ method itself may also begin to explain some of the uncertainty, e.g., anomalously strong non-$CO_2$ forcing coinciding with anomalously strong warming (see Supplementary of Burton et al. (2023) for $FCO_2$ on SST in the six models considered in this paper).